# Exploring the Algorithm-Dependent Generalization of AUPRC Optimization with List Stability

**Peisong Wen**[1,2]  **Qianqian Xu**[1]*  **Zhiyong Yang**[2]
**Yuan He**[3]  **Qingming Huang**[1,2,4,5*]

[1] Key Lab of Intell. Info. Process., Inst. of Comput. Tech., CAS
[2] School of Computer Science and Tech., University of Chinese Academy of Sciences
[3] Alibaba Group
[4] BDKM, University of Chinese Academy of Sciences
[5] Peng Cheng Laboratory
{wenpeisong20z,xuqianqian}@ict.ac.cn
{yangzhiyong21,qmhuang}@ucas.ac.cn    heyuan.hy@alibaba-inc.com

## Abstract

Stochastic optimization of the Area Under the Precision-Recall Curve (AUPRC) is a crucial problem for machine learning. Although various algorithms have been extensively studied for AUPRC optimization, the generalization is only guaranteed in the multi-query case. In this work, we present the first trial in the single-query generalization of stochastic AUPRC optimization. For sharper generalization bounds, we focus on algorithm-dependent generalization. There are both algorithmic and theoretical obstacles to our destination. From an algorithmic perspective, we notice that the majority of existing stochastic estimators are biased when the sampling strategy is biased, and is leave-one-out unstable due to the non-decomposability. To address these issues, we propose a sampling-rate-invariant unbiased stochastic estimator with superior stability. On top of this, the AUPRC optimization is formulated as a composition optimization problem, and a stochastic algorithm is proposed to solve this problem. From a theoretical perspective, standard techniques of the algorithm-dependent generalization analysis cannot be directly applied to such a listwise compositional optimization problem. To fill this gap, we extend the model stability from instancewise losses to listwise losses and bridge the corresponding generalization and stability. Additionally, we construct state transition matrices to describe the recurrence of the stability, and simplify calculations by matrix spectrum. Practically, experimental results on three image retrieval datasets on speak to the effectiveness and soundness of our framework.

## 1 Introduction

Area Under the Precision-Recall Curve (AUPRC) is a widely used metric in the machine learning community, especially in learning to rank, which effectively measures the trade-off between precision and recall of a ranking model. Compared with threshold-specified metrics like accuracy and recall@k, AUPRC reflects a more comprehensive performance by capturing all possible thresholds. In addition, literature has shown that AUPRC is insensitive toward data distributions [20], making it adaptable to largely skewed data. Benefiting from these appealing properties, AUPRC has become one of the standard metrics in various applications, *e.g.*, retrieval [55, 58, 22, 41], object detection [45, 49, 15], medical diagnosis [50, 35], and recommendation systems [16, 73, 1, 64, 2].

Over the past decades, the importance of AUPRC has prompted extensive researches on direct AUPRC optimization. Early work focuses on full-batch optimization [45, 44, 26]. However, in

---

*Corresponding authors.

the era of deep learning, the rapidly growing scale of models and data makes these full-batch algorithms infeasible. Therefore, in recent years, it has raised an increasing favor of the stochastic AUPRC optimization [9, 12, 31, 46]. Since AUPRC optimization is a stochastic dependent compositional optimization problem, general convergence rates are infeasible for AUPRC optimization. To fill this gap, [54, 70, 69] provide AUPRC optimization algorithms with provable convergence. See Appendix A for more on related work.

Despite the promoting performance of these methods in various scenarios, the generalization of AUPRC optimization algorithms is still an open problem. Some studies [17, 63] provide provable generalization for AUPRC optimization in information retrieval. In this scene, a dataset consists of multiple queries, where each query corresponds to a set of positive and negative samples. However, these results require sufficient queries to ensure small generalization errors, but leave the single-query case alone, *i.e.*, *whether the generalization error tends to zero with the length of a single-query increasing is still unclear*. This limits the adaptation scope of these methods. To fill this gap, in this paper we aim to ***design a stochastic optimization framework for AUPRC with a provable algorithm-dependent generalization performance in the single-query case***.

The target is challenging in three aspects: **(a)** Most AUPRC stochastic estimators are biased with a biased sampling rate. Moreover, due to the non-decomposability, outputs of existing algorithms might change a lot with slight changes in the training data, which is called *leave-one-out unstable* in this paper. Such an unstability is harmful to the generalization. **(b)** The standard framework to analyze the algorithm-dependent generalization requires the objective function to be expressed as a sum of instancewise terms, while AUPRC involves a listwise loss. **(c)** The stochastic optimization of AUPRC is a two-level compositional optimization problem, which brings more complicated proofs of the stability.

In search of a solution to **(a)**, we propose a sampling-rate-invariant asymptotically unbiased stochastic estimator based on a reformulation of AUPRC. Notably, to ensure the stability [28, 39, 37, 38] of the estimator, the objective is formulated as a two-level compositional problem by introducing an auxiliary vector for the ranking estimation. Error analysis further supports the feasibility of our method, and inspires us to add a semi-variance regularization term. To solve this problem, we propose an algorithm with provable convergence that combines stochastic gradient descent (SGD), linear interpolation and exponential moving average.

Facing challenge **(b)**, we extend instancewise model stability to listwise model stability, and correspondingly put forward the generalization via stability of listwise problems. On top of this, we bridge the generalization of AUPRC and the stability of the proposed optimization algorithm.

As for challenge **(c)**, since the variables to be optimized are typically updated alternately in the compositional optimization problem, we propose state transition matrices of these variables, and simplify the calculations of the stability with matrix spectrum.

In a nutshell, the main contributions of this paper are summarized as follows:

- Algorithmically, a stochastic learning algorithm is proposed for AUPRC optimization. The core of the proposed algorithm is a stochastic estimator which is sampling-rate-invariant asymptotically unbiased.

- Theoretically, we present the first trial on the algorithm-dependent generalization of stochastic AUPRC optimization. To the best of our knowledge, it is also the first work to analyze the stability of stochastic compositional optimization problems.

- Technically, we extend the concept of the stability and generalization guarantee to listwise non-convex losses. Then we simplify the stability analysis of compositional objective by matrix spectrum. These techniques might be instructive for other complicated metrics.

## 2 Problem Formulation

### 2.1 Preliminaries on AUPRC

**Notations.** Consider a set of $N$ examples $\mathcal{S} = \{(\boldsymbol{x}_i, y_i)\}_{i=1}^N$ independently drawn from a sample space $\mathcal{D} = \mathcal{X} \times \mathcal{Y}$, where $\mathcal{X}$ is the input space and $\mathcal{Y} = \{-1, 1\}$ is the label space. For sake of the presentation, denote the set of positive examples of $\mathcal{S}$ as $\mathcal{S}^+ = \{\boldsymbol{x}_i^+\}_{i=1}^{N^+}$, and similarly the set

of negative examples is denoted as $\mathcal{S}^- = \{\boldsymbol{x}_i^-\}_{i=1}^{N^-}$, where $N^+ = |\mathcal{S}^+|, N^- = |\mathcal{S}^-|$. With a slight abuse of notation, we also denote $\mathcal{S} = \mathcal{S}^+ \cup \mathcal{S}^-$ if there is no ambiguity. Generally, we assume that the dataset is sufficiently large, such that $N^+/(N^+ + N^-) = \mathbb{P}(y = 1) := \pi$. Our target is to learn a score function $h_{\boldsymbol{w}} : \mathcal{X} \mapsto \mathbb{R}$ with parameters $\boldsymbol{w} \in \Omega \subseteq \mathbb{R}^d$, such that the scores of positive examples are higher than negative examples. Furthermore, when appling the score function to a dataset $\mathcal{S} \in \mathcal{X}^N$, we denote $h_{\boldsymbol{w}} : \mathcal{X}^N \mapsto \mathbb{R}^N$, where the $k$-th element of $h_{\boldsymbol{w}}(\mathcal{S})$ has the top-$k$ values of $\{h_{\boldsymbol{w}}(\boldsymbol{x})|\boldsymbol{x} \in \mathcal{S}\}$. Denote the asymptotic upper bound on complexity as $\mathcal{O}$, and denote asymptotically equivalent as $\asymp$.

In this work, our main interest is to optimize a score function in the view of AUPRC:

$$
\begin{aligned}
\mathrm{AUPRC}(\boldsymbol{w}; \mathcal{D}) &= \int_0^1 \mathbb{P}(y = 1|h_{\boldsymbol{w}}(\boldsymbol{x}) \geq c) \, d\, \mathbb{P}(h_{\boldsymbol{w}}(\boldsymbol{x}) \geq c|y = 1) \\
&= \int_0^1 \frac{\pi TPR(c)}{\pi TPR(c) + (1-\pi)FPR(c)} \, d\, \mathbb{P}(h_{\boldsymbol{w}}(\boldsymbol{x}) \geq c|y = 1),
\end{aligned}
\tag{1}
$$

where $(\boldsymbol{x}, y) \sim \mathcal{D}$, $c$ refers to a threshold, and $TPR(c) = \mathbb{P}(h_{\boldsymbol{w}}(\boldsymbol{x}) \geq c|y = 1), FPR(c) = \mathbb{P}(h_{\boldsymbol{w}}(\boldsymbol{x}) \geq c|y = 0)$. For a finite set $\mathcal{S}$, AUPRC is typically approximated by replacing the distribution function $\mathbb{P}(h_{\boldsymbol{w}}(\boldsymbol{x}) \geq c|y = 1)$ with its empirical cumulative distribution function [8, 19]:

$$
\widehat{\mathrm{AUPRC}}(\boldsymbol{w}; \mathcal{S}) = \hat{\mathbb{E}}_{\boldsymbol{x}^+ \sim \mathcal{S}^+} \left[ \frac{\pi \widehat{TPR}(h_{\boldsymbol{w}}(\boldsymbol{x}^+))}{\pi \widehat{TPR}(h_{\boldsymbol{w}}(\boldsymbol{x}^+)) + (1-\pi)\widehat{FPR}(h_{\boldsymbol{w}}(\boldsymbol{x}^+))} \right],
\tag{2}
$$

where $\widehat{TPR}(c) = \hat{\mathbb{E}}_{\boldsymbol{x} \sim \mathcal{S}^+}[\ell_{0,1}(c - h_{\boldsymbol{w}}(\boldsymbol{x}))], \widehat{FPR}(c) = \hat{\mathbb{E}}_{\boldsymbol{x} \sim \mathcal{S}^-}[\ell_{0,1}(c - h_{\boldsymbol{w}}(\boldsymbol{x}))], \ell_{0,1}(x) = 1$ if $x \leq 0$ or $\ell_{0,1}(x) = 0$ otherwise. It has been shown that $\widehat{\mathrm{AUPRC}}$ is an unbiased estimator when $N^+/(N^+ + N^-) \to \pi$ and $N \to \infty$ [8]. With the above estimation, we have the following optimization objective:

$$
\min_{\boldsymbol{w}} \quad \widehat{\mathrm{AUPRC}}^{\downarrow}(\boldsymbol{w}; \mathcal{S}) = 1 - \widehat{\mathrm{AUPRC}}(\boldsymbol{w}; \mathcal{S}) = \hat{\mathbb{E}}_{\boldsymbol{x}^+ \sim \mathcal{S}^+} \left[ \sigma \left( \frac{1-\pi}{\pi} \cdot \frac{\widehat{FPR}(h_w(\boldsymbol{x}^+))}{\widehat{TPR}(h_w(\boldsymbol{x}^+))} \right) \right],
\tag{3}
$$

where $\sigma(x) = x/(1 + x)$ is concave and monotonically increasing. To make it smooth, surrogate losses $\ell_1, \ell_2$ are used to replace $\ell_{0,1}$ in $\widehat{FPR}$ and $\widehat{TPR}$ respectively, yielding the following surrogate objective:

$$
\min_{\boldsymbol{w}} \quad f(\boldsymbol{w}; \mathcal{S}) = \hat{\mathbb{E}}_{\boldsymbol{x}^+ \sim \mathcal{S}^+} \left[ \sigma \left( \frac{1-\pi}{\pi} \cdot \frac{\widehat{FPR}(h_w(\boldsymbol{x}^+); \ell_1)}{\widehat{TPR}(h_w(\boldsymbol{x}^+); \ell_2)} \right) \right],
\tag{4}
$$

where $\widehat{TPR}(c; \ell_2) = \hat{\mathbb{E}}_{\boldsymbol{x} \sim \mathcal{S}^+}[\ell_2(c - h_{\boldsymbol{w}}(\boldsymbol{x}))], \widehat{FPR}(c; \ell_1) = \hat{\mathbb{E}}_{\boldsymbol{x} \sim \mathcal{S}^-}[\ell_1(c - h_{\boldsymbol{w}}(\boldsymbol{x}))]$. Specifically, when $N^+/(N^+ + N^-) = \pi$, it is equivalent to another commonly used formulation **Average Precision (AP) Loss**:

$$
\widehat{\mathrm{AP}}^{\downarrow}(\boldsymbol{w}; \mathcal{S}) = \hat{\mathbb{E}}_{\boldsymbol{x}^+ \sim \mathcal{S}^+} \left[ \sigma \left( \frac{\sum_{\boldsymbol{x} \sim \mathcal{S}^-}[\ell_1(h_{\boldsymbol{w}}(\boldsymbol{x}^+) - h_{\boldsymbol{w}}(\boldsymbol{x}))]}{\sum_{\boldsymbol{x} \sim \mathcal{S}^+}[\ell_2(h_{\boldsymbol{w}}(\boldsymbol{x}^+) - h_{\boldsymbol{w}}(\boldsymbol{x}))]} \right) \right].
\tag{5}
$$

## 2.2 Stochastic Learning of AUPRC

Under the stochastic learning framework for instancewise losses, the empirical risk $F(\boldsymbol{w}; \mathcal{S})$ is expressed as a sum of instancewise losses: $F(\boldsymbol{w}; \mathcal{S}) = \frac{1}{N} \sum_{\boldsymbol{x} \sim \mathcal{S}} \hat{f}(\boldsymbol{w}; \boldsymbol{x})$, where $\hat{f}(\boldsymbol{w}; \boldsymbol{x})$ is the stochastic estimator of $F(\boldsymbol{w}; \mathcal{S})$. Different from instancewise losses, listwise losses like AUPRC require a batch of samples to calculate the stochastic estimator. Specifically, at each step, a subset of $\mathcal{S}$: $\boldsymbol{z} = \boldsymbol{z}^+ \cup \boldsymbol{z}^-$ is randomly drawn, where $\boldsymbol{z}^+$ consists of $n^+$ positive examples and $\boldsymbol{z}^-$ consists of $n^-$ negative examples. Then a stochastic estimator of the loss function, denoted as $\hat{f}(\boldsymbol{w}; \boldsymbol{z})$, is computed with $\boldsymbol{z}$. Similar to the instancewise case, we consider a variant of the empirical/population AUPRC risks as approximations, which is a sum of stochastic losses w.r.t. all posible $\boldsymbol{z}$:

$$
F(\boldsymbol{w}; \mathcal{S}) = \frac{1}{M} \sum_{\boldsymbol{z}} \hat{f}(\boldsymbol{w}; \boldsymbol{z}), \quad F(\boldsymbol{w}) = \mathbb{E}_{\mathcal{S} \sim \mathcal{D}}[F(\boldsymbol{w}; \mathcal{S})],
\tag{6}
$$

where $M$ is the number of all posible $\boldsymbol{z}$. Unfortunately, due to the non-decomposability of the empirical AUPRC risk $f(\boldsymbol{w}; \mathcal{S})$, it is tackle to determine the approximation errors between $F(\boldsymbol{w}; \mathcal{S})$ and

$f(\boldsymbol{w}; \mathcal{S})$ in general. Nonetheless, in Sec. 3.3 we argue that by selecting proper $\hat{f}(\boldsymbol{w}; \boldsymbol{z})$, $F(\boldsymbol{w}; \mathcal{S})$ can be asymptotically unbiased estimator of $f(\boldsymbol{w}; \mathcal{S})$, which naturally makes $F(\boldsymbol{w})$ an asymptotically unbiased estimator of $1 - \text{AUPRC}$. **In this case, $\hat{f}$ is said to be an asymptotically unbiased stochastic estimator. Moreover, if the unbiasedness holds under biased sampling rate, it is said to be sampling-rate-invariant asymptotically unbiased.**

# 3 Asymptotically Unbiased Stochastic AUPRC Optimization

In this section, we will present our SGD-style stochastic optimization algorithm of AUPRC. In Sec. 3.1, we propose surrogate losses to make the objective function differentiable. In Sec. 3.2, we present details of the proposed stochastic estimator and the corresponding optimization algorithm. Analyses on approximation errors are provided in Sec. 3.3.

## 3.1 Differentiable Surrogate Losses

Since $\ell_{0,1}$ appears in both the numerator and denominator of Eq. (4), simply implementing $\ell_1, \ell_2$ with a single function [56, 9, 54] will bring difficulty to analyze the relationship between $\widehat{\text{AUPRC}}^{\downarrow}(\boldsymbol{w}; \mathcal{S})$ and $f(\boldsymbol{w}; \mathcal{S})$. This motivates us to choose $\ell_1 \geq \ell_{0,1}, \ell_2 \leq \ell_{0,1}$, such that $\widehat{\text{AUPRC}}^{\downarrow}(\boldsymbol{w}; \mathcal{S}) \leq f(\boldsymbol{w}; \mathcal{S})$, thus the original empirical risk could be optimized by minimizing its upper bound $f(\boldsymbol{w}; \mathcal{S})$. Concretely, $\ell_1$ and $\ell_2$ are defined as the one-side Huber loss and the one-side sigmoid loss:

$$\ell_1(x) = \begin{cases} -2x/\tau_1, & x < 0, \\ (1 - x/\tau_1)^2, & 0 \leq x < \tau_1, \\ 0, & x \geq \tau_1. \end{cases} \qquad \ell_2(x) = \begin{cases} \dfrac{\exp(-x/\tau_2) - 1}{\exp(-x/\tau_2) + 1}, & x < 0, \\ 0, & x \geq 0. \end{cases} \tag{7}$$

Here $\tau_1, \tau_2 > 0$ are hyperparameters. $\ell_1$ is convex and decreasing, which ensures the gap between positive-negative pairs is effectively optimized. Additionally, compared with the square loss and the exponential loss, $\ell_1$ is more robust to noises. $\ell_2$ is Lipschitz continuous, and $\ell_2 \to \ell_{0,1}$ with $\tau_2 \to 0$.

## 3.2 Stochastic Estimator of AUPRC

The key to a stochastic learning framework is the design of the stochastic estimator (or the corresponding gradients), i.e., $\hat{f}(\boldsymbol{w}; \boldsymbol{z})$. Existing methods [9, 74, 12] implement it with $\widehat{\text{AP}}^{\downarrow}(\boldsymbol{w}; \boldsymbol{z})$ (Eq. (5)), which might suffer from two problems:

**(P1)** Comparing Eq. (4) and Eq. (5), it can be seen that only when $n^+/(n^+ + n^-) \to \pi$, $\widehat{\text{AP}}^{\downarrow}$ is an asymptotically unbiased estimator. However, it is hardly satisfied since the sampling strategy is usually biased in practice.

**(P2)** Each term in the summation of $\widehat{\text{AP}}^{\downarrow}$ is related to all instances of a batch, leading to weak leave-one-out stability, i.e., changing one instance might result in a relatively large fluctuation in the stochastic gradient, especially when changing a positive example.

To tackle the above problems, we first substitute $\widehat{FPR}(h_w(\boldsymbol{x}^+); \ell_1)$ with $\hat{\mathbb{E}}_{\boldsymbol{x} \sim \boldsymbol{z}^-}[\ell_1(h_{\boldsymbol{w}}(\boldsymbol{x}^+) - h_{\boldsymbol{w}}(\boldsymbol{x}))]$, and then introduce an auxiliary vector $\boldsymbol{v} \in \mathbb{R}^{N^+}$ to estimate $\widehat{TPR}$. Formally, we propose the following batch-based estimator:

$$\hat{f}(\boldsymbol{w}; \boldsymbol{z}) = \hat{f}(\boldsymbol{w}; \boldsymbol{z}, \boldsymbol{v}) = \hat{\mathbb{E}}_{\boldsymbol{x}^+ \sim \boldsymbol{z}^+} \left[ \sigma \left( \frac{1 - \pi}{\pi} \cdot \frac{\hat{\mathbb{E}}_{\boldsymbol{x} \sim \boldsymbol{z}^-}[\ell_1(h_{\boldsymbol{w}}(\boldsymbol{x}^+) - h_{\boldsymbol{w}}(\boldsymbol{x}))]}{\hat{\mathbb{E}}_{v \sim \boldsymbol{v}}[\ell_2(h_{\boldsymbol{w}}(\boldsymbol{x}^+) - v)]} \right) \right]. \tag{8}$$

Such an estimator enjoys two advantages: in terms of **P1**, it is asymptotically unbiased regardless of the sampling rate (see Sec. 3.3 for detailed discussions); as for **P2**, we use $\boldsymbol{v}$ to substitute $h_{\boldsymbol{w}}(\mathcal{S}^+)$, such that each positive example in a mini-batch only appears in one term. Ideally, it can be considered as using all positive examples in the dataset to estimate $\widehat{TPR}$ instead of that from a mini-batch. With the fact that $n^- \gg n^+$, this makes the corresponding algorithm more stable. Moreover, based on the model stability, generalization bounds are available (see Sec. 4).

### 3.3 Analyses on Approximation Errors

In this subsection, we analyze errors from two approximations in the above algorithm: **1)** the gap between $F(\boldsymbol{w}; \mathcal{S})$ and the true AUPRC loss; **2)** the gap between the interpolated scores $\phi(h_{\boldsymbol{w}}(\boldsymbol{z}^+))$ and the true scores $h_{\boldsymbol{w}}(\mathcal{S}^+)$. Proofs are provided in Appendix B.1.

Denote $\pi = N^+/(N^+ + N^-)$ and $\pi_0 = n^+/(n^+ + n^-)$. We would like to show that for all $\boldsymbol{w} \in \Omega$, $\mathbb{E}_{\boldsymbol{z}}[\hat{f}(\boldsymbol{w}; \boldsymbol{z})]$ is an unbiased estimator when $n \to \infty$, no matter how $\pi_0$ is chosen, while for $\mathbb{E}_{\boldsymbol{z}}[\widehat{\mathrm{AP}}^{\downarrow}(\boldsymbol{w}; \boldsymbol{z})]$, it holds only when $\pi_0 = \pi$. Since only one model $\boldsymbol{w}$ is considered, we let $\boldsymbol{w}_t = \boldsymbol{w}$ in the update rule of $\boldsymbol{v}$ (Eq. (10)), and we have the following proposition:

**Proposition 1.** *Consider updating $\boldsymbol{v}$ with Eq. (10) for $T$ steps, then we have*

$$\mathbb{E}[\boldsymbol{v}] = \mathbb{E}[\phi(h_{\boldsymbol{w}}(\boldsymbol{z}^+))] + (1 - \beta)^T \left(\boldsymbol{v}_1 - \mathbb{E}[\phi\left(h_{\boldsymbol{w}}(\boldsymbol{z}^+)\right)]\right), \quad Var[\boldsymbol{v}] \leq Var[\phi(h_{\boldsymbol{w}}(\boldsymbol{z}^+))] \cdot \frac{\beta}{2 - \beta}.$$

**Remark 1.** *Two conclusions could be drawn from the above proposition: first, if the linear interpolation is asymptotically unbiased (see next subsection), by choosing a large $T$ or setting $\boldsymbol{v}_1 = \mathbb{E}[\phi(h_{\boldsymbol{w}}(\boldsymbol{z}^+))]$, we have $\mathbb{E}[\boldsymbol{v}] \approx h_{\boldsymbol{w}}(\mathcal{S}^+)$; second, **by choosing a smaller $\beta$, $\boldsymbol{v}$ is more likely to concentrate on $h_{\boldsymbol{w}}(\mathcal{S}^+)$**.*

**Proposition 2.** *Assume the linear interpolation is asymptotically unbiased. Let $\kappa_1^2 = \hat{\mathbb{E}}_{c \sim h_{\boldsymbol{w}}(\boldsymbol{z}^+)}[Var_{\boldsymbol{x} \sim \mathcal{S}^-}[\ell_1(c - h_{\boldsymbol{w}}(\boldsymbol{x}))]]$, $\kappa_2^2 = \hat{\mathbb{E}}_{c \sim h_{\boldsymbol{w}}(\boldsymbol{z}^+)}[Var_{v \sim \boldsymbol{v}}[\ell_2(c - v)]]$. When $\kappa_1^2/n^- \to 0$, $\kappa_2^2/n^+ \to 0$, then there exists a positive scale $H$, such that*

$$\hat{\mathbb{E}}_{\boldsymbol{z} \subseteq \mathcal{S}}[\hat{f}(\boldsymbol{w}; \boldsymbol{z})] \xrightarrow{P} \widehat{AUPRC}^{\downarrow}(\boldsymbol{w}; \mathcal{S}), \quad \hat{\mathbb{E}}_{\boldsymbol{z} \subseteq \mathcal{S}}\left[\widehat{AP}^{\downarrow}(\boldsymbol{w}; \boldsymbol{z})\right] \xrightarrow{P} (1 + (\pi_0 - \pi)H) \cdot \widehat{AUPRC}^{\downarrow}(\boldsymbol{w}; \mathcal{S}),$$

*where $\xrightarrow{P}$ refers to convergence in probability, and $\boldsymbol{z} \subseteq \mathcal{S}$ refers to subsets described in Sec. 2.2.*

**Remark 2.** *The above proposition suggests that the proposed batch-based estimator is sampling-rate-invariant asymptotically unbiased, while $\widehat{AP}^{\downarrow}$ tends to be larger when the sampling rate of the positive class is greater than the prior, and vice versa. We also provide a non-asymptotic result in Appendix B.2.*

Simulation experiments are conducted as complementary to the theory. Following previous work [8], the scores are drawn from three types of distributions, including binormal, bibeta and offset uniform. The results of binormal distribution are visualized in Fig. 1, and detailed descriptions and more results are available in Appendix B.2. These results are consistent with the above remark.

Next we further study the interpolation error. For the sake of presentation, denote $p : [0, 1] \mapsto \mathbb{R}$ to be an increasing score function describing $h_{\boldsymbol{w}}(\mathcal{S}^+)$, where $p(x)$ is the score in the bottom $x$-quantile of $h_{\boldsymbol{w}}(\mathcal{S}^+)$. Similarly, let $\hat{p}$ to be the interpolation results of $\mathbb{E}_A[h_{\boldsymbol{w}}(\boldsymbol{z}^+)]$. Assume that $\mathbb{E}_A[h_{\boldsymbol{w}}(\boldsymbol{z}^+)]$ are located in the $(i/n^+)$-quantiles of $p$, where $i \in [n^+]$, such that $p(i/n^+) = \hat{p}(i/n^+)$ and all interpolation intervals are with length $1/n^+$. The following proposition provides an upper bound of the approximation error (see [61] for proof):

**Proposition 3** (**Linear Interpolation Error**). *Let $p, \hat{p}$ be defined as above. Then we have*

$$\|p - \hat{p}\|_\infty \leq \|p''\|_\infty / \left(8(n^+)^2\right).$$

Similar to the last subsection, simulation results are shown in Fig. 1(c), which shows the expected errors of linear interpolation are ignorable.

### 3.4 Optimization Algorithm

In the rest of this section, we focus on how to optimize $F(\boldsymbol{w}; \mathcal{S})$. The main challenge is to design update rules for $\boldsymbol{v}$, such that it could efficiently and effectively approximate $h_{\boldsymbol{w}}(\mathcal{S}^+)$ without full-batch scanning. To overcome the challenge, we propose an algorithm called **Stochastic Optimization of AUPRC (SOPRC)**, which jointly updates model parameters $\boldsymbol{w}$ and the auxiliary vector $\boldsymbol{v}$. A summary of the detailed process is shown as Alg. 1. At step $t$, a batch of data is sampled from the training set, and then compute the corresponding scores. Afterward, scores of positive examples are mapped into a $N^+$-dimension vector with linear interpolation $\phi$ as shown in Alg. 2. $\boldsymbol{v}_{t+1}$ are updated with the interpolated scores in a moving average manner.

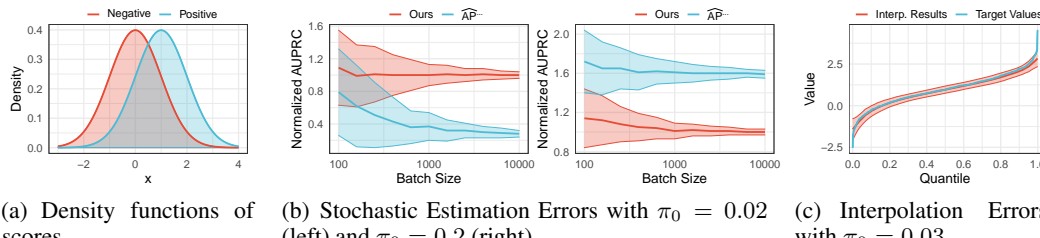

(a) Density functions of scores.

(b) Stochastic Estimation Errors with $\pi_0 = 0.02$ (left) and $\pi_0 = 0.2$ (right).

(c) Interpolation Errors with $\pi_0 = 0.03$.

Figure 1: Empirical analysis of estimation errors on simulation data.

Practically, $n^+, n^-$ are finite, causing inevitable estimation errors in $f(\boldsymbol{w}; \boldsymbol{z}_{i_t}, \boldsymbol{v}_{t+1})$. Notice that another factor influencing the stochastic estimation errors, *i.e.*, $\kappa_1^2$ and $\kappa_2^2$. To reduce them, it is expected that the variance of positive (negative) scores are small, which motivates us to add a variance regularization term. However, it might force to reduce positive scores that higher than the mean value, which is contrary to our target. Therefore, we propose a **semi-variance regularization term** [4]:

$$\mathcal{L}_{var} = \frac{\lambda_1}{n^+} \sum_{\substack{\boldsymbol{x} \sim \boldsymbol{z}^+ \\ h_{\boldsymbol{w}}(\boldsymbol{x}) < \mu^+}} (h_{\boldsymbol{w}}(\boldsymbol{x}) - \mu^+)^2 + \frac{\lambda_2}{n^-} \sum_{\substack{\boldsymbol{x} \sim \boldsymbol{z}^- \\ h_{\boldsymbol{w}}(\boldsymbol{x}) > \mu^-}} (h_{\boldsymbol{w}}(\boldsymbol{x}) - \mu^-)^2, \qquad (9)$$

where $\mu^+ = \frac{1}{n^+} \sum_{\boldsymbol{x} \sim \boldsymbol{z}^+} h_{\boldsymbol{w}}(\boldsymbol{x})$, $\mu^- = \frac{1}{n^-} \sum_{\boldsymbol{x} \sim \boldsymbol{z}^-} h_{\boldsymbol{w}}(\boldsymbol{x})$, $\lambda_1, \lambda_2$ are hyperparameters. Finally, we compute the gradients of $f(\boldsymbol{w}; \boldsymbol{z}_{i_t}, \boldsymbol{v}_{t+1}) + \mathcal{L}_{var}$, and update parameters $\boldsymbol{w}$ with gradient descent.

---

**Algorithm 1** SOPRC

**Input:** Training dataset $\mathcal{S}$, maximum iterations $T$, learning rate $\{\eta_t\}_{t=1}^T$ and $\{\beta_t\}_{t=1}^T$.
**Output:** model parameters $\boldsymbol{w}_{T+1}$.
1: Initialize model parameters $\boldsymbol{w}_1$ and $\boldsymbol{v}_1$.
2: **for** $t = 1$ to $T$ **do**
3:     Sample a subset $\boldsymbol{z}_{i_t}$ from $\mathcal{S}$.
4:     Compute $h_{\boldsymbol{w}_t}(\boldsymbol{z}_{i_t}^+)$ and map the results into $\phi(h_{\boldsymbol{w}_t}(\boldsymbol{z}_{i_t}^+))$ with Alg. 2.
5:     Update $\boldsymbol{v}$ with
$$\boldsymbol{v}_{t+1} = (1 - \beta_t)\boldsymbol{v}_t \\ + \beta_t \phi(h_{\boldsymbol{w}_t}(\boldsymbol{z}_{i_t}^+)). \qquad (10)$$
6:     Compute $\mathcal{L}_{var}$ with Eq. (9).
7:     Update the model parameter:
$$\boldsymbol{w}_{t+1} = \boldsymbol{w}_t - \eta_t \cdot \nabla \mathcal{L}_{var} \\ - \eta_t \cdot \nabla f(\boldsymbol{w}_t; \boldsymbol{z}_{i_t}, \boldsymbol{v}_{t+1}). \qquad (11)$$
8: **end for**

---

**Algorithm 2** Score Interpolation $\phi(\cdot)$

**Input:** A real value vector $\boldsymbol{u} \in \mathbb{R}^n$ where $n < N^+$, range of target values $[b, B]$.
**Output:** Interpolated vector $\boldsymbol{m} = \phi(\boldsymbol{u})$.
1: Sort $\boldsymbol{u}$ in descending order.
2: Initialize $\boldsymbol{m}$ as $\boldsymbol{0}_{N^+}$, let $u_0 = max(2u_1 - u_2, b)$, $u_{n+1} = min(2u_n - 2u_{n-1}, B)$.
3: **for** $i = 1$ to $n$ **do**
4:     **for** $j = \lceil \frac{N^+ \cdot (i-1)}{n} \rceil$ to $\left[ \frac{N^+ \cdot i}{n} \right]$ **do**
5:         $m_j += \left[ (i - jn/N^+)u_{i-1} \\ + (1 + jn/N^+ - i)u_i \right]/2$
6:     **end for**
7:     **for** $j = \left[ \frac{N^+ \cdot i}{n} \right]$ to $\lfloor \frac{N^+ \cdot (i+1)}{n} \rfloor$ **do**
8:         $m_j += \left[ (i + 1 - jn/N^+)u_{i-1} \\ + (jn/N^+ - i)u_i \right]/2$
9:     **end for**
10: **end for**

---

## 4 Generalization of SOPRC via Stability

In this section, we turn to study the *excess generalization error* of the proposed algorithm. Formally, following standard settings [5], we consider the test error of the model $A(\mathcal{S})$ trained on the training set $\mathcal{S}$. Our target is to seek an upper bound of the excess error $\mathbb{E}_{A,\mathcal{S}}[F(A(\mathcal{S})) - F(\boldsymbol{w}^*)]$, where $\boldsymbol{w}^* \in \arg\min_{\boldsymbol{w} \in \Omega} \mathbb{E}_{A,\mathcal{S}}[F(\boldsymbol{w}^*)]$. It can be decomposed as:

$$\mathbb{E}_{\mathcal{S},A}[F(A(\mathcal{S})) - F(\boldsymbol{w}^*)] = \underbrace{\mathbb{E}_{\mathcal{S},A}[F(A(\mathcal{S})) - F(A(\mathcal{S}); \mathcal{S})]}_{\textit{Estimation Error}} + \underbrace{\mathbb{E}_{\mathcal{S},A}[F(A(\mathcal{S}); \mathcal{S}) - F(\boldsymbol{w}^*)]}_{\textit{Optimization Error}}.$$

The estimation error sources from the gap of minimizing the empirical risk instead of the expected risk. In Sec. 4.1, we provide detailed discussion on the estimation error. The optimization error measures the gap between the minimum empirical risk and the results obtained by the optimization

algorithm, which will be studied in Sec. 4.2. Detailed proofs of this section are available in Appendix C. Before the formal presentation, we show the main assumptions:

**Assumption 1** (**Bounded Scores & Gradient**). $|\hat{f}(\boldsymbol{w};\cdot)| \leq B, \|\nabla \hat{f}(\boldsymbol{w};\cdot)\|_2 \leq G$ *for all* $\boldsymbol{w} \in \Omega$.

**Assumption 2** (**L-Smooth Loss**). $\|\nabla \hat{f}(\boldsymbol{w};\cdot) - \nabla \hat{f}(\tilde{\boldsymbol{w}};\cdot)\|_2 \leq L\|\boldsymbol{w} - \tilde{\boldsymbol{w}}\|_2$ *for all* $\boldsymbol{w}, \tilde{\boldsymbol{w}} \in \Omega$.

**Assumption 3** (**Lipschitz Continuous Functions**). $|\ell_1(x) - \ell_1(\tilde{x})| \leq L_1|x - \tilde{x}|, |\ell_2(x) - \ell_2(\tilde{x})| \leq L_2|x - \tilde{x}|$ *for all* $x, \tilde{x} \in [-2B, 2B]$. $\|\phi(\boldsymbol{x}) - \phi(\tilde{\boldsymbol{x}})\|_2 \leq C_\phi\|\boldsymbol{x} - \tilde{\boldsymbol{x}}\|_2$ *for all* $\boldsymbol{x}, \tilde{\boldsymbol{x}} \in \mathbb{R}^{N^+}$.

### 4.1 Generalization of AUPRC via Model Stability

The generalization of SGD-style algorithms for instancewise loss has been widely studied with stability measure [39, 21, 28]. However, these results could not be directly applied to listwise losses like AUPRC. The main reason is that the estimation of each stochastic gradient requires a list of examples, and the estimation is usually biased. Nonetheless, to bridge the optimization algorithm and the generalization of AUPRC, we propose a listwise variant of *on-average model stability* [39] as follows:

**Definition 1** (**Listwise On-average Model Stability**). *Let* $\mathcal{S} = \{(\boldsymbol{x}_i, y_i)\}_{i=1}^N$ *and* $\widetilde{\mathcal{S}} = \{(\widetilde{\boldsymbol{x}}_i, y_i)\}_{i=1}^N$ *be two sets of examples whose features are drawn independently from* $\mathcal{X}$. *For any* $i = 1, \cdots, N$, *denote* $\mathcal{S}^{(i)} = \{(\boldsymbol{x}_1, y_1), \cdots, (\boldsymbol{x}_{i-1}, y_{i-1}), (\widetilde{\boldsymbol{x}}_i, y_i), (\boldsymbol{x}_{i+1}, y_{i+1}), \cdots, (\boldsymbol{x}_n, y_n)\}$. *A stochastic algorithm* $A$ *is listwise on-average model* $(\epsilon^+, \epsilon^-)$-*stable if the following condition holds:*

$$\mathbb{E}_{\mathcal{S},\widetilde{\mathcal{S}},A}\left[\frac{1}{N^+}\sum_{y_i=1}\left\| A(\mathcal{S}) - A(\mathcal{S}^{(i)})\right\|_2\right] \leq \epsilon^+, \mathbb{E}_{\mathcal{S},\widetilde{\mathcal{S}},A}\left[\frac{1}{N^-}\sum_{y_i=-1}\left\| A(\mathcal{S}) - A(\mathcal{S}^{(i)})\right\|_2\right] \leq \epsilon^-.$$

The following theorem shows that the estimation error is bounded by the above-defined stability:

**Theorem 1** (**Generalization via Model Stability**). *Let a stochastic algorithm* $A$ *be listwise on-average model* $(\epsilon^+, \epsilon^-)$-*stable and Asmp.* 1 *holds. Then we have*

$$\mathbb{E}_{\mathcal{S},A}\left[F(A(\mathcal{S})) - F(A(\mathcal{S});\mathcal{S})\right] \leq G(n^+\epsilon^+ + n^-\epsilon^-). \tag{12}$$

With the above theorem, now we only need to focus on the model stability of the proposed algorithm. Notice that in Alg. 1, both $\boldsymbol{w}_t$ and $\boldsymbol{v}_t$ are updated at each step, thus we have to consider the stability of both simultaneously. The following lemma provides a recurrence for the stability $\boldsymbol{w}_t$ and $\boldsymbol{v}_t$.

**Lemma 1.** *Let* $\mathcal{S}, \widetilde{\mathcal{S}}, \mathcal{S}^{(i)}$ *be constructed as Def.* 1 *and Asmp.* 1, 2, 3 *hold. Let* $\{\boldsymbol{w}_t\}_t$ *and* $\{\boldsymbol{w}_t^{(i)}\}_t$ *be produced by Alg.* 1 *with* $\mathcal{S}$ *and* $\mathcal{S}^{(i)}$, *respectively. Denote* $L = \max\{L_w, L_v/n^+, C_\phi B, G/2, B'_\ell\}$, $\boldsymbol{m}_t^{(i)} = \left[\begin{array}{ccc} \|\boldsymbol{w}_t - \boldsymbol{w}_t^{(i)}\|_2 & \|\boldsymbol{v}_t - \boldsymbol{v}_t^{(i)}\|_2 & 1 \end{array}\right]^\top$, $\boldsymbol{m}_t^+ = \frac{1}{N^+}\sum_{y_i=1}\mathbb{E}_{\mathcal{S},A}\left[\boldsymbol{m}_t^{(i)}\right]$, $\boldsymbol{m}_t^- = \frac{1}{N^+}\sum_{y_i=-1}\mathbb{E}_{\mathcal{S},A}\left[\boldsymbol{m}_t^{(i)}\right]$. *Then for all* $t \in [T]$, *by setting* $\beta_t \leq 2C_\phi B/n^+$, *we have*

$$\boldsymbol{m}_{t+1}^+ \leq \left(\boldsymbol{I}_3 + \boldsymbol{R}_t^+\right) \cdot \boldsymbol{m}_t^+, \quad \boldsymbol{m}_{t+1}^- \leq \left(\boldsymbol{I}_3 + \boldsymbol{R}_t^-\right) \cdot \boldsymbol{m}_t^-, \tag{13}$$

*where* $\boldsymbol{I}_3$ *is the* $3 \times 3$ *identity matrix and*

$$R_t^+ = \left[\begin{array}{ccc} 2L\eta_t & \frac{L(1-\beta_t)\eta_t}{N^+} & \frac{L\eta_t}{N^+} \\ L\beta_t & 0 & \frac{1}{N^+} \\ 0 & 0 & 0 \end{array}\right], R_t^- = \left[\begin{array}{ccc} 2L\eta_t & \frac{L_v(1-\beta_t)\eta_t}{N^+} & \frac{L\eta_t \cdot n^+}{N^-} \\ L\beta_t & 0 & 0 \\ 0 & 0 & 0 \end{array}\right]. \tag{14}$$

Finally, we utilize the matrix spectrum of $R_t^+$ and $R_t^-$ to show that the model stability w.r.t. Alg. 1 decreases as the number of training examples increases (see Appendix C.2 for details):

**Theorem 2.** *Let* $\lambda = LC_\eta(1 + \sqrt{1 - \beta^2 + \beta})$, *and assumptions in Lem.* 1 *hold. By setting* $\eta_t \leq \frac{C_\eta}{t}$, $\beta_t = \beta \asymp 1/n^+$ *and* $T \leq N^+$, *Alg.* 1 *is list on-average model stable with*

$$\epsilon^+ = \mathcal{O}\left(\frac{(Tn^+)^{\frac{\lambda}{\lambda+1}}}{N^+}\right), \epsilon^- = \mathcal{O}\left(\frac{(Tn^-)^{\frac{\lambda}{\lambda+1}}}{N^-}\right). \tag{15}$$

Table 1: Quantitative results on SOP, iNaturalist, and VehicleID. All methods are trained with training sets. The best and the second best results are highlighted in soft red and soft blue, respectively.

| Methods | Stanford Online Products | | | iNaturalist | | | PKU VehicleID | | |
|---|---|---|---|---|---|---|---|---|---|
| | mAUPRC | R@1 | R@10 | mAUPRC | R@1 | R@4 | mAUPRC | R@1 | R@5 |
| Contrastive loss [27] | 57.73 | 77.60 | 89.31 | 27.99 | 54.19 | 71.12 | 67.26 | 87.46 | 94.60 |
| Triplet loss [32] | 58.07 | 78.34 | 90.50 | 30.59 | 60.53 | 77.62 | 70.99 | 90.09 | 95.54 |
| MS loss [71] | 60.10 | 79.64 | 90.38 | 30.28 | 63.39 | 78.50 | 69.15 | 88.82 | 95.06 |
| XBM [72] | 61.29 | 80.66 | 91.08 | 27.46 | 59.12 | 75.18 | 71.24 | 92.78 | 95.83 |
| SmoothAP [9] | 61.65 | 81.13 | 92.02 | 33.92 | 66.13 | 80.93 | 72.28 | 91.31 | 96.05 |
| DIR [58] | 60.74 | 80.52 | 91.35 | 33.51 | 64.86 | 79.79 | 72.72 | 91.38 | 96.10 |
| FastAP [12] | 57.10 | 77.30 | 89.61 | 31.02 | 56.64 | 73.57 | 70.82 | 89.42 | 95.38 |
| AUROC [25] | 55.80 | 77.32 | 89.64 | 27.24 | 60.88 | 77.76 | 58.12 | 81.73 | 91.92 |
| BlackBox [52] | 59.74 | 79.48 | 90.74 | 29.28 | 56.88 | 74.10 | 70.92 | 90.14 | 95.52 |
| Ours | 62.75 | 81.91 | 92.50 | 36.16 | 68.22 | 82.86 | 74.92 | 92.56 | 96.43 |

## 4.2 Convergence of AUPRC Stochastic Optimization

Following previous work [24, 34], we study the optimization error of the proposed algorithm under the *Polyak-Łojasiewicz (PL)* condition. It has been shown that the PL condition holds for several widely used models including some classes of neural networks [13, 42].

**Assumption 4 (Polyak-Łojasiewicz Condition [34, 37]).** *Denote $\boldsymbol{w}^* = \arg\min_{\boldsymbol{w}\in\Omega} F(\boldsymbol{w})$. Assume $F$ satisfy the expectation version of PL condition with parameter $\mu > 0$, i.e.,*

$$\mathbb{E}_{\mathcal{S}}[F(\boldsymbol{w};\mathcal{S}) - F(\boldsymbol{w}^*)] \leq \frac{1}{\mu}\mathbb{E}_{\mathcal{S}}[\|\nabla F(\boldsymbol{w};\mathcal{S})\|_2^2]. \tag{16}$$

The main difference to the existing convergence analysis on non-convex optimization is that the gradient estimation is biased. Nonetheless, we show that the bias terms from Alg. 1 tend to 0 with sufficient training data and training time (see Appendix C.3), leading to the following convergence:

**Theorem 3.** *Let Asmp. 1, 3, 4 hold. By setting $\eta_t = \frac{2t+1}{\mu(t+1)^2}$ and $\beta_t = \beta \asymp 1/n^+$, we have*

$$\mathbb{E}_A[F(\boldsymbol{w}_{T+1}) - F(\boldsymbol{w}^*)] = \mathcal{O}\left(n^+/T + 1/N^+\right). \tag{17}$$

**Theorem 4.** *Let assumptions in Thm. 2 and 3 hold. By setting $T \asymp (N^+)^{\frac{\lambda+1}{2\lambda+1}}(n^+)^{-\frac{1}{2\lambda+1}}$, we have*

$$\mathbb{E}_{\mathcal{S},A}[F(A(\mathcal{S})) - F(\boldsymbol{w}^*)] = \mathcal{O}\left((N^+)^{-\frac{\lambda+1}{2\lambda+1}} \cdot (n^+)^{\frac{3\lambda+1}{2\lambda+1}}\right) + \mathcal{O}\left((N^-)^{-\frac{\lambda+1}{2\lambda+1}} \cdot (n^-)^{\frac{3\lambda+1}{2\lambda+1}}\right). \tag{18}$$

**Remark 3.** *Recall that $\lambda = LC_\eta(1 + \sqrt{1 - \beta^2 + \beta})$ and $C_\eta = 4/\mu$, when $\beta$ is small, we have $\lambda \approx 4L/\mu$. Here $L/\mu$ is a condition number determined by the model and surrogate losses. Notice that $n^+ \ll N^+, n^- \ll N^-$, if $\lambda = 1$, the generalization bound is $\mathcal{O}\left((N^+)^{-2/3} \cdot (n^+)^{4/3} + (N^-)^{-2/3} \cdot (n^-)^{4/3}\right)$. As $\lambda$ increases, it increases to $\mathcal{O}\left((N^+)^{-1/2} \cdot (n^+)^{3/2} + (N^-)^{-1/2} \cdot (n^-)^{3/2}\right)$.*

## 5 Experiments

To validate the effectiveness of the proposed method, we conduct empirical studies on the image retrieval task, in which data distributions are largely skewed and AUPRC is commonly used as an evaluation metric. More detailed experimental settings are provided in Appendix D.1. The source code is available in https://github.com/KID-7391/SOPRC.git.

### 5.1 Implementation Details

**Datasets.** We evaluate the proposed method on three image retrieval benchmarks with various domains and scales, including **Stanford Online Products (SOP)**[2] [48], **PKU VehicleID**[3] [43] and

---

[2]https://github.com/rksltnl/Deep-Metric-Learning-CVPR16. Licensed MIT.

[3]https://www.pkuml.org/resources/pku-vehicleid.html. Data files © Original Authors.

**iNaturalist**[4] [68]. We follow the official setting to split a test set from each dataset, and then further split the rest into a training set and a validation set by a ratio of $9:1$.

**Network Architecture.** The feature extractor is implemented with ResNet-50 [29] pretrained on ImageNet [60]. Following previous work [12, 9], the batch normalization layers are fixed during training, and the output embeddings are mapped to 512-d with a linear projection. Given $L_2$ normalized embeddings of a query image $e_q$ and a gallery list $\{e_i\}_i$, the scores are represented by the cosine similarity $e_q^\top e_i$ for all $i$.

**Optimization Strategy.** In the training phase, the input images are resized such that the sizes of the shorter sides are 256. Afterward, we applied standard data augmentations including random cropping ($224 \times 224$) and random flipping ($50\%$). The model parameters are optimized in an end-to-end manner as shown in Alg. 1, where $\beta = 0.001$ and the weight decay is set to $4 \times 10^{-4}$. The default batch size is set to 224, where each mini-batch is randomly sampled such that there are exactly 4 positive examples per category. The learning rates are tuned according to performance on validation sets: for SOP, the learning rate are initialized as $0.01$ and decays by $0.1$ at the $15k$ and $30k$ iterations, $T = 50k$; for VehicleID, the learning rate are initialized as $0.001$ and decays by $0.1$ at the $40k$ and $80k$ iterations, $T = 100k$; for iNaturalist, the learning rate are initialized as $0.001$ and decays by $0.1$ at the $80k$ and $110k$ iterations, $T = 130k$.

**Competitors.** We compare two types of competitors: **1) Pairwise Losses**, including *Contrastive Loss* [27], *Triplet Loss* [32], *Multi-Similarity (MS) Loss* [71], *Cross-Batch Memory (XBM)* [72]. These methods construct loss functions with image pairs or triplets. **2) Ranking-Based Losses**, including *SmoothAP* [9], *FastAP* [12], *DIR* [58], *BlackBox* [33], and *Area Under the ROC Curve Loss (AUROC)* [76]. These methods directly optimize the ranking-based metrics.

**Evaluation Metrics.** In all experiments, we adopt evaluation metrics: *mean AUPRC (mAUPRC)* and *Recall@k*. mAUPRC is also called mean average precision (mAP) in literature, which takes the mean value over the AUPRC of all queries. Recall@k measures the probability that at least one positive example is ranked in the top-k list.

## 5.2 Main Results

We evaluate all methods with *mean AUPRC (mAUPRC)* and *Recall@k*. mAUPRC measures the mean value of the AUPRC over all queries, a.k.a. mean average precision (mAP). The performance comparisons on test sets are shown in Tab. 1. Consequently, we have the following observations: **1)** In all datasets, the proposed method surpasses all competitors in the view of mAUPRC, especially in the large-scale long-tailed dataset iNaturalist. This validates the advantages of our method in boosting the AUPRC of models. **2)** Compared to pairwise losses, the AUPRC/AP optimization methods enjoy better performance generally. The main reason is that pairwise losses could only optimize models indirectly by constraining relative scores between positive and negative example pairs, while ignoring the overall ranking. **3)** Although some pairwise methods like XBM have a satisfying performance on Recall@1, their mAUPRC is relatively low. It is caused by the limitation of Recall@1, *i.e.*, it focuses on the top-1 score while ignoring the ranking of other examples. What's more, this phenomenon shows the inconsistency of Recall@k and AUPRC, revealing the necessity of studying AUPRC optimization. More results are available in Appendix D.2.

To qualitatively demonstrate the effect of the proposed method, we also show the mean PR curves and convergence curves in Fig. 2. The left two subfigures demonstrate that the proposed method achieves can effectively improve AUPRC. The right subfigure shows the affect of batch size, from which can be seen that a large batch size leads to better performance. One of the reasons is that a small batch size will amplify the AUPRC stochastic estimation error. Such a problem have been addressed by maintaining inner gradient estimations [54, 70, 75]. Unfortunately, when applied to image retrieval problems, it needs to maintain intermediate variables for each positive pair, bringing high complexity in time and space, thus we leave this problem as further work.

## 5.3 Ablation Studies

We further investigate the effect of different components of the proposed method. Results are shown in Tab. 2, and more detailed statements and analyses are as follows.

---

[4] https://github.com/visipedia/inatcomp/tree/master/2018. Licensed MIT.

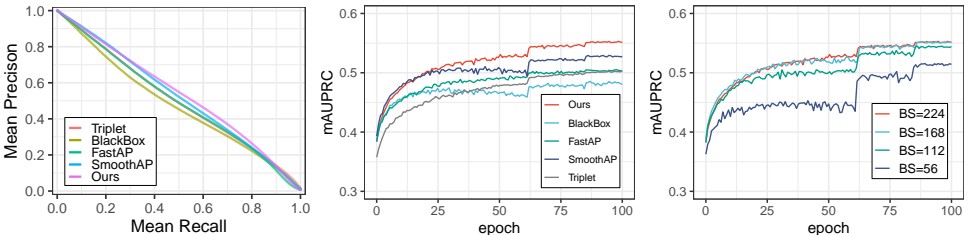

Figure 2: Qualitative results on iNaturalist. Left most: mean PR curves of different methods. Right two: convergence of different methods and batch sizes in terms of mAUPRC in the validation set.

Table 2: Ablation study over different components of our method on iNaturalist.

| No. | Unb. Est. | with $v_t$ | with $\mathcal{L}_{var}$ | Opt. | mAUPRC | R@1 | R@4 | R@16 | R@32 |
|-----|-----------|------------|-------------------------|------|--------|------|------|-------|-------|
| 1 | ✗ | ✗ | ✗ | SGD | 34.58 | 66.35 | 81.04 | 89.80 | 92.72 |
| 2 | ✓ | ✗ | ✗ | SGD | 35.84 | 67.08 | 81.68 | 90.17 | 92.98 |
| 3 | ✓ | ✓ | ✗ | SGD | 35.99 | 67.50 | 82.03 | 90.44 | 93.26 |
| 4 | ✓ | ✓ | ✓ | SGD | 36.16 | 68.22 | 82.86 | 91.02 | 93.71 |
| 5 | ✓ | ✓ | ✓ | Adam | 36.20 | 68.48 | 82.70 | 90.96 | 93.63 |

**Effect of Unbiased Estimator.** To show the performance drop caused by the biased estimator, we replace the prior $\pi$ in Eq. (8) with $n^+/(n^+ + n^-)$. Comparing line 1 and line 2, using the unbiased estimator increases the mAUPRC by 1.3%, which is consistent with our theoretical results in Sec. 3.3. Notably, the unbiased estimator is the main source of improvements in terms of mAUPRC.

**Effect of $v_t$.** To show the effect of introducing $v_t$ to estimate $\phi(\mathcal{S}^+)$, we directly use $\phi(z^+)$ instead in the first two lines. Comparing line 2 and line 3, using $v_t$ could bring consistent improvements due to the better generalization ability.

**Effect of $\mathcal{L}_{var}$.** We show that shrinking variances could reduce the batch-based estimation errors. Comparing line 3 and line 4, it can be seen that $\mathcal{L}_{var}$ further boosts the proposed method.

**Effect of Optimizer.** Comparing line 4 and line 5, it can be seen that the choice of optimizer only has a slight influence.

## 6 Conclusion & Future Work

In this paper, we present a stochastic learning framework for AUPRC optimization. To begin with, we propose a stochastic AUPRC optimization algorithm based on an asymptotically unbiased stochastic estimator. By introducing an auxiliary vector to approximate the scores of positive examples, the proposed algorithm is more stable. On top of this, we study algorithm-dependent generalization. First, we propose list model stability to handle listwise losses like AUPRC, and bridge the generalization and the stability. Afterward, we show that the proposed algorithm is stable, leading to an upper bound of the generalization error. Experiments on three benchmarks validate the advantages of the proposed framework. One limitation is the convergence rate is controlled by the scale of the dataset. In the further, we will consider techniques like variance reduction to improve the convergence rate, and jointly consider the corresponding algorithm-dependent generalization.

## Acknowledgments

This work was supported in part by the National Key R&D Program of China under Grant 2018AAA0102000, in part by National Natural Science Foundation of China: U21B2038, 61931008, 6212200758 and 61976202, in part by the Fundamental Research Funds for the Central Universities, in part by Youth Innovation Promotion Association CAS, in part by the Strategic Priority Research Program of Chinese Academy of Sciences, Grant No. XDB28000000, in part by the China National Postdoctoral Program for Innovative Talents under Grant BX2021298, and in part by China Postdoctoral Science Foundation under Grant 2022M713101.

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
