# OpenReview forum: "Exploring the Algorithm-Dependent Generalization of AUPRC Optimization with List Stability"
_NeurIPS.cc/2022/Conference — NeurIPS 2022 Accept_

### Official Review · Reviewer_8mrY · 2022-07-10

**Rating:** 7
**Confidence:** 5
**Soundness:** 4 excellent
**Presentation:** 3 good
**Contribution:** 4 excellent

**Summary:**

In this paper, the authors aim to explore the properties of the AUPRC stochastic optimization. They provide two main theoretical results: the unbiasedness of stochastic estimators and the generalization of the optimization algorithm. To develop the algorithm-dependent generalization, they extend the model stability to list-wise loss. Experiments are conducted on three datasets.

**Questions:**

1. In section 3.2 (P1), it is claimed that $n^+ / (n^+ + n^-) = \pi$ is hard to satisfied in practice. I think this condition can be achieved by changing the sampling rate during training, could you provide more explanations?
2. In my view, the $O(1/N^+)$ term is caused by the estimation of $h(S^+)$. Is it possible to use acceleration techniques in optimization, like variance reduction, to avoid this problem?

**Limitations:**

Yes

**Strengths And Weaknesses:**

The theoretical results of this paper are solid in the following aspects:
1. It’s interesting and challenging to study the algorithm-dependent generalization of AUPRC optimization. This work fills the gap in the list-wise loss stability analysis.
2. The authors provide a useful tool to analyze the convergence/stability of compositional optimization problems. Although transition matrices are widely used in complicated processes, it is novel and effective to simplify the calculation with the matrix spectrum.
3. By jointly considering the convergence and the generalization, this work provides guidance to find the trade-off between convergence and generalization of AUPRC optimization.
4. The main results are well presented and clearly proved.

My main concern is the presentation of some key techniques can be further improved. While the proposed techniques like simplification with matrix spectrum sound novel and reasonable to me, they haven’t been formally present in the main paper. Presenting it briefly in the main paper allows researchers to reuse these techniques on other problems.

Overall, this paper addresses some important theoretical issues of AUPRC optimization, and I tend to accept this paper.

---

> ### Author Response · Authors · 2022-08-01
> **Response To Reviewer 8mrY**
>
> Thank you for your time and constructive feedback! We will improve our work as outlined below:
>
> ### **Q1:**
> Formal presentation of the simplification technique with matrix spectrum.
>
> ### **A1:**
> The simplification technique is used in the proof of Thm. 2 (from Eq. (63) to Eq. (67)). To optimize a compositional problem
> $$
>     \min\_{\pmb{w}} f(\pmb{w}, g(\pmb{w})),
> $$
> a commonly used technique is maintaining an intermediate variable $\pmb{v} \approx g(\pmb{w})$. Consider two datasets $\mathcal{S},\mathcal{S}'$ that differ with at most one example. Let $\pmb{w}\_t, \pmb{v}\_t$ to be the model and the intermediate variable generated with the dataset $\mathcal{S}$ respectively, and similarly $\pmb{w}\_t', \pmb{v}\_t'$ are generated with $\mathcal{S}'$.
> When analyzing the corresponding stability, we have to bound both $\\|\pmb{w}\_t' - \pmb{w}\_t\\|$ and $\\|\pmb{v}\_t' - \pmb{v}\_t\\|$. Since the two variables depend on each other, the derivation of the upper bound will be cumbersome. We propose to solve this problem from the recurrence. Formally, if
> $$
>     \left[\\|\pmb{w}\_{t+1}' - \pmb{w}\_{t+1}\\|, \\|\pmb{v}\_{t+1}' - \pmb{v}\_{t+1}\\|, 1\right]^\top
>     \leq \left(\pmb{I}\_3 + \pmb{M} / t\right) \left[\\|\pmb{w}\_{t}' - \pmb{w}\_{t}\\|, \\|\pmb{v}\_{t}' - \pmb{v}\_{t}\\|, 1\right]^\top, \\
>     \\|\pmb{w}\_{t\_0}' - \pmb{w}\_{t\_0}\\| = \\|\pmb{v}\_{t\_0}' - \pmb{v}\_{t\_0}\\| = 0,
> $$
> where all elements in $\pmb{M}$ is non-negative, then we have
> $$
>     \\|\pmb{w}\_{T+1}' - \pmb{w}\_{T+1}\\|
>     \leq [1\ 0\ 0]\Lambda\ diag\left((T')^{\lambda\_1},(T')^{\lambda\_2},(T')^{\lambda\_3}\right)\Lambda^{-1}\ [0\ 0\ 1]^{\top},
> $$
> where $T' = T / (t\_0-1)$, $\lambda\_{1,2,3}$ are the eigenvalues of $M$, and each column of $\Lambda$ is the corresponding eigenvector. In this way, the stability for compositional optimization algorithms can be obtained by analyzing the spectrum of transition matrices.
>
> ### **Q2:**
> Why is the condition $n^{+} / (n^{+} + n^{-}) = \pi$ hard to satisfied in practice?
>
> ### **A2:**
> The sampling strategy depends on the specific tasks, and the assumption of the sampling rate might limit the generality of the AUPRC optimization algorithm. Here we provide two examples:
>
> **Example 1.** Data distributions in some tasks like retrieval and medical diagnosis are largely skewed, e.g., for class \#2588 in iNaturalist, $\pi \approx 1.2\times 10^{-4}$. Satisfying the condition $n^{+} / (n^{+} + n^{-}) = \pi$ requires a batch-size of over $10\times 10^{4}$, which is neither feasible nor necessary, especially for deep models.
>
> **Example 2.** We first briefly introduce a common sampling setting in retrieval tasks: a mini-batch usually contains multiple queries and corresponding positive examples. Given a query, the negative examples are formed from positive examples of other queries. When the priors of queries are different, it will be hard to control the number of examples to satisfy the condition for all queries.
>
> ### **Q3:**
> Is it possible to use techniques like variance reduction to avoid the $\mathcal{O}(1/N^{+})$ term in Thm. 3?
>
> ### **A3:**
> The $\mathcal{O}(1 / N^{+})$ term sources from the non-linearity of $\nabla f(\pmb{w}\_t;\pmb{z}\_{i\_t}, \cdot)$ (see Eq. (72), Eq. (79) and Lem. 4). Therefore, even if $\phi(h\_{\pmb{w}}(\pmb{z}^+))$ is an unbiased estimation of $h\_{\pmb{w}}(\mathcal{S}^+)$, the stochastic gradient might still be biased. Recent work on bilevel optimization has utilized variance reduction to handle similar problems, so it is feasible to solve the issue with similar techniques. However, it will make the stability analysis much more complicated, and we have to explore this in future work.

---

### Official Review · Reviewer_RxCi · 2022-07-11

**Rating:** 7
**Confidence:** 4
**Soundness:** 4 excellent
**Presentation:** 3 good
**Contribution:** 3 good

**Summary:**

This work proposes a stochastic algorithm for AUPRC optimization based on a sampling-rate-invariant unbiased stochastic estimator. The authors study the theoretical properties including the approximation error and generalization bound. Based on the theoretical results, they propose a semi-variance regular term to further improve the performance. The algorithm is applied to the image retrieval task.

**Questions:**

Please see the weaknesses for my main concerns. Some minor issues are given as follows:
- The performance gain is more significant in iNaturalist than SOP. What’s the difference between these two datasets?
- How to determine the prior $\pi$ in the image retrieval task?


**Limitations:**

 The limitations and potential negative societal impact have been described.

**Strengths And Weaknesses:**

Strengths:
- The weaknesses of some previous works are clearly identified and the proposed method is tailored to alleviate them. Both theoretical analysis and simulation experiments are provided to support the soundness.
- Formal theoretical interpretation of the components of the proposed method. This may inspire future work along this line.
- The extensive experiments have validated the proposed method. The ablation studies clearly present the effect of each component.

Weaknesses:
- The experiment details of the competitors need to be more specific to ensure fair comparisons.
- How the surrogate loss $\ell_1$ is chosen seems unclear. The theoretical results hold as long as it is an upper bound of $\ell_{0,1}$, why not use commonly used surrogate losses like square loss?

---

> ### Author Response · Authors · 2022-08-01
> **Response To Reviewer RxCi**
>
> We sincerely thank you for your time and efforts! We would like to clarify the following issues:
>
> ### **Q1:**
> Experiment details of the competitors.
>
> ### **A1:**
> We reimplement all the competitors on the same codebase to ensure a consistent setting in terms of model structure, data preprocessing and augmentation, learning rate schedule, testing pipeline, etc. The unique hyperparameters of competitors follow the optimal settings of the original papers. Moreover, the optimizers used are slightly different: following previous work, Adam is used to train the competitors, while ours is trained with SGD to ensure consistency with our theoretical analysis. We also provide the results of ours trained with Adam in Tab. 2, which shows no significant difference. Therefore, the comparisons are fair, and we will update these details in the latest version.
>
>
> ### **Q2:**
> The reason for choosing the one-side Huber loss as a surrogate loss.
>
> ### **A2:**
> On one hand, one-side Huber loss can be viewed as a smooth variant of the Hinge loss, such that Assumption 2 (i.e., the objective function $F$ is $L$-smooth) holds. On the other hand, compared to the square loss, one-side Huber loss is less sensitive to outliers, making the learning more robust.
>
>
> ### **Q3:**
> Why the performance gain is more significant in iNaturalist than SOP?
> ### **A3:**
> In fact, the performance gain of all AUPRC-based methods (e.g., SmoothAP [9], DIR [57], FastAP [12]) is more significant in iNaturalist than SOP (See Tab. 1). As far as we know, the main reason is two-fold: first, the number of positive examples per query is much larger ($56.7$ in iNaturalist v.s. $5.3$ in SOP on average). Therefore, according to the definition of AUPRC, the weights of positive examples should be more discriminative in datasets like iNaturalist, while pairwise losses like contrastive loss ignore this factor. Second, the scale of iNaturalist is larger, thus it is less possible to overfit the training set of iNaturalist than SOP.
>
>
> ### **Q4:**
> How to determine the prior $\pi$ in the image retrieval task?
>
> ### **A4:**
> We count the number of examples in each class and estimate the prior $\pi$ with the frequency. Assuming that the training set is i.i.d. sampled from the true distribution, such an estimation of $\pi$ is unbiased and consistent.

---

### Official Review · Reviewer_qm2w · 2022-07-11

**Rating:** 8
**Confidence:** 4
**Soundness:** 4 excellent
**Presentation:** 4 excellent
**Contribution:** 3 good

**Summary:**

The paper proposes a novel framework that optimizes AUPRC in an end-to-end manner. The main idea is inspired by the theoretical properties of the objective function and the stochastic algorithm. The authors show that the objective function is asymptotically unbiased by approximation error analysis, and the proposed stochastic optimization algorithm has a generalization guarantee. Their experiments demonstrate the proposed framework work well on image retrieval datasets.

**Questions:**

I hope the authors can clarify some more in-depth analysis mentioned in the weaknesses part. Besides, the stochastic estimator is only proved to be asymptotically unbiased, but the order is still unclear. I suggest the authors to provides the asymptotic order. It might require data distribution hypotheses, as what the authors did in simulation experiments.

**Limitations:**

Yes.

**Strengths And Weaknesses:**

I consider this work novel and sound in three aspects:
1) The proposed stochastic estimator solves the estimation bias issue. The instability is mitigated by an auxiliary vector estimating positive scores. Sufficient theories and numerical experiments illustrate the proposed method.
2) It is the first work studying the algorithm-dependent generalization of stochastic AUPRC optimization. It is a challenging topic involving the list-wise problem and compositional problem.
3) The main claims are properly verified on both simulation and real-work data.

The main weakness of this work is missing explanations/analyzes of some techniques:
1) In table 2, the model with the semi-variance term has higher R@K, but similar mAUPRC. I’m interested in the mechanism of the semi-variance regulation.
2) The time consumption of the proposed score interpolation is $O(N^+)$. It might slow down the training process.

To sum up, this paper looks solid on both technical and theoretical parts. The presentation is overall well organized. Therefore, I recommend accepting this paper.

---

> ### Author Response · Authors · 2022-08-01
> **Proof of Proposition 3**
>
> In the proof of Prop. 2, we denote
> $$
>     X^c\_{{n^-}} = \hat{\mathbb{E}}\_{\pmb{x}\sim\pmb{z}^-}\left[\ell\_1\left(c - h\_{\pmb{w}}(\pmb{x})\right)\right],~~~~
>     Y^c\_{{n^+}} = \hat{\mathbb{E}}\_{v\sim\pmb{v}}\left[\ell\_2\left(c - v\right)\right].
> $$
> Then the approximation error is decomposed into
> $$
> \begin{aligned}
>     &\mathop{\hat{\mathbb{E}}}\limits\_{\pmb{z}\subseteq\mathcal{S}}[\hat{f}(\pmb{w};\pmb{z})] - \widehat{\text{AUPRC}}^\downarrow(\pmb{w};\mathcal{S}) \\\\
>     =& \underbrace{\mathop{\hat{\mathbb{E}}}\limits\_{\pmb{z}, c\sim h\_{\pmb{w}}(\pmb{z}^+)}\left[
>         \frac{(1-\pi) X^c\_{{n^-}}}{(1-\pi) X^c\_{{n^-}} + \pi Y^c\_{{n^+}}} - \frac{(1-\pi) X^c\_{{n^-}}}{(1-\pi) \mu\_{c,1} + \pi Y^c\_{{n^+}}}
>     \right]}\_{(a)} \\\\
>     &+ \underbrace{\mathop{\hat{\mathbb{E}}}\limits\_{\pmb{z}, c\sim h\_{\pmb{w}}(\pmb{z}^+)}\left[\frac{(1-\pi) X^c\_{{n^-}}}{(1-\pi) \mu\_{c,1} + \pi \mu\_{c,2}} - \frac{(1-\pi) \mu\_{c,1}}{(1-\pi) \mu\_{c,1} + \pi \mu\_{c,2}}
>     \right]}\_{(b)} \\\\
>     &+ \underbrace{\mathop{\hat{\mathbb{E}}}\limits\_{\pmb{z}, c\sim h\_{\pmb{w}}(\pmb{z}^+)}\left[\frac{(1-\pi) X^c\_{{n^-}}}{(1-\pi) \mu\_{c,1} + \pi Y^c\_{{n^+}}} - \frac{(1-\pi) X^c\_{{n^-}}}{(1-\pi) \mu\_{c,1} + \pi \mu\_{c,2}}
>     \right]}\_{(c)},
> \end{aligned}
> $$
>
> where $\mu\_{c,1},\mu\_{c,2}$ are the mean values of $X^c\_{{n^-}}$ and $Y^c\_{{n^+}}$, respectively.
>
> Next, we focus on the term $(a)$. Under the assumption of Prop. 2, $X^c\_{{n^-}}$ can be viewed as an average of i.i.d. variables, thus according to Hoeffding's inequality, for any $\epsilon > 0$ we have
> $$
>     \mathbb{P}\left(\left|X^c\_{{n^-}} - \mu\_{c,1}\right| \geq \epsilon\right) \leq 2\exp\left(-\frac{2{n^-} \epsilon^2}{B\_{\ell\_1}^2}\right).
> $$
> Therefore, for any $c$, $0 < \delta < 1$, with probability at least $1 - \delta$,  we have
> $$
> \begin{aligned}
>     &\left|\frac{(1-\pi) X^c\_{{n^-}}}{(1-\pi) X^c\_{{n^-}} + \pi Y^c\_{{n^+}}} - \frac{(1-\pi) X^c\_{{n^-}}}{(1-\pi) \mu\_{c,1} + \pi Y^c\_{{n^+}}}\right| \\\\
>     \leq& \left|\frac{(1-\pi)^2 X^c\_{{n^-}}}{\left((1-\pi) X^c\_{{n^-}} + \pi Y^c\_{{n^+}}\right)\left((1-\pi) \mu\_{c,1} + \pi Y^c\_{{n^+}}\right)}\right|\cdot \left|X^c\_{{n^-}} - \mu\_{c,1}\right| \\\\
>     \leq& \frac{1}{\mu\_{c,1}} \cdot \left|X^c\_{{n^-}} - \mu\_{c,1}\right| \\\\
>     \leq& \sqrt{\frac{B\_{\ell\_1}^2 \log \frac{2}{\delta}}{2\mu\_{c,1}^2 {n^-}}}\\\\
>     \leq& \sqrt{\frac{B\_{\ell\_1}^2 \log \frac{2}{\delta}}{2\mu\_{1}^2 {n^-}}},
> \end{aligned}
> $$
> where $\mu\_1 = \inf\_{c}\ \mu\_{c,1}$. If we further assume that $X^c\_{{n^-}}$ is independent w.r.t. different $c$, then considering all positive $c$, with probability at least $1 - \delta / 3$, we have
> $$
>     |(a)| \leq \sqrt{\frac{B\_{\ell\_1}^2 \log \frac{6{n^+}}{\delta}}{2\mu\_{1}^2 {n^-}}} = \mathcal{O}\left(\sqrt{\frac{\log (6{n^+} / \delta)}{{n^-}}}\right),
> $$
> and similarly
> $$
>     |(b)| = \mathcal{O}\left(\frac{\log (6{n^+} / \delta)}{{n^-}}\right), ~~ |(c)| = \mathcal{O}\left(\sqrt{\frac{\log (6{n^+} / \delta)}{{n^+}}}\right).
> $$
> To sum up, with probability at least $1 - \delta$ we have
> $$
>     \left|\mathop{\hat{\mathbb{E}}}\limits\_{\pmb{z}\subseteq\mathcal{S}}[\hat{f}(\pmb{w};\pmb{z})] - \widehat{\text{AUPRC}}^\downarrow(\pmb{w};\mathcal{S})\right| \leq |(a)| + |(b)| + |(c)| = \mathcal{O}\left(\sqrt{\frac{\log (6{n^+} / \delta)}{{n^+}}} + 2\sqrt{\frac{\log (6{n^+} / \delta)}{{n^-}}}\right).
> $$

---

> ### Author Response · Authors · 2022-08-01
> **Response To Reviewer qm2w**
>
> Thank you for your time and positive comments on our manuscript! We would like to reply to the following questions:
> ### **Q1:**
> The mechanism of the semi-variance regulation. Why the improvement in R@K is more significant than mAUPRC?
> ### **A1:**
> The semi-variance regulation is motivated by Prop. 2, which says that reducing the score variance will lead to better AUPRC estimation. To explore the effect of semi-variance regulation, we report the standard deviation of positive/negative scores in the validation set of iNaturalist:
>
> |    method    | pos std  | neg std  |  mAUPRC  |  R@1  |
> | :------------: |   :----:   |   :----:   |   :----:   | :-----: |
> | w   semi-var |   0.10   |   0.05   |  36.16   | 68.22 |
> | w/o semi-var |   0.14   |   0.08   |  35.99   | 67.50 |
>
> It can be seen that the semi-variance regulation significantly reduces the standard deviation, and improves both mAUPRC and R@1. Since R@K only considers whether there are positive examples in the top-K list, while AUPRC requires a better overall ranking. Therefore, it is more challenging to improve AUPRC, and it seems that the improvement will be less significant in absolute value.
>
>
> ### **Q2:**
> The time consumption of the proposed score interpolation.
>
> ### **A2:**
> Although the score interpolation is shown as a two-loop process in Alg. 2, obviously it can be accelerated with a parallel implementation. Practically, in our Pytorch implementation, it takes $2.9ms$ to compute the AUPRC loss per iteration, of which only $0.4ms$ is used for the score interpolation. Compared to the time spent on model inference and update (about $526ms$ per iteration), the time consumption of loss calculation and score interpolation is negligible.
>
> ### **Q3:**
> What's the non-asymptotic approximation error of the stochastic estimator under specific distribution hypotheses?
>
> ### **A3:**
> Thank you for your helpful suggestions! Besides the asymptotic analysis in our paper, we still have the following non-asymptotic conclusion on the approximation error:
>
> **Proposition 3.** *For any $0 < \delta < 1$, at least with probability of $1 - \delta$, we have*
> $$
> \left|\mathop{\hat{\mathbb{E}}}\limits\_{\pmb{z}\subseteq\mathcal{S}}[\hat{f}(\pmb{w};\pmb{z})] - \widehat{\text{AUPRC}}^\downarrow(\pmb{w};\mathcal{S})\right| = \mathcal{O}\left(\sqrt{\frac{\log (6{n^+} / \delta)}{{n^+}}} + 2\sqrt{\frac{\log (6{n^+} / \delta)}{{n^-}}}\right).
> $$
> **The proof is provided in the next comment.**
>
> From the above Proposition, we can draw the conclusion that the approximation error convergences to zero at order of $\mathcal{O}\left(\frac{1}{\sqrt{{n^+}}} + \frac{2}{\sqrt{{n^-}}}\right)$. However, such non-asymptotic analysis leaves the variance out, while the asymptotic version explains why the semi-variance term works (see A1 to Reviewer qm2w).

---

### Meta-Review · Area_Chair_JNZh · 2022-08-25

**Recommendation:** Accept
**Confidence:** Certain

**Metareview:**

All reviewers agree the paper makes novel contributions for AUPRC optimization.

It proposes new batch-based estimator of AUPRC and studies its approximation error. Then it develops a new algorithms for optimizing this estimator. It also establishes the generalization error of the proposed algorithms via novel listwise stability.

It seems that the proposed method is still sensitive to the batch size as shown in the results. The authors are encouraged to compare with [53], which proposes a stochastic algorithm for AP maximization with convergence guarantee and is not sensitive to the batch size.

**Award:**

No

---

### Decision · Program_Chairs · 2022-09-14

Accept